# Unintended pregnancy and contraception use among African women living with HIV: Baseline analysis of the multi-country US PEPFAR PROMOTE cohort

Jim Aizire[1]*, Nonhlanhla Yende-Zuma[2,3], Sherika Hanley[4], Teacler Nematadzira[5], Mandisa M. Nyati[6], Sufia Dadabhai[1], Lameck Chinula[7,8], Catherine Nakaye[9], Mary Glenn Fowler[10], Taha Taha[1], for the US-PEPFAR PROMOTE Cohort Study team[¶]

1 Department of Epidemiology, Johns Hopkins Bloomberg School of Public Health, Baltimore, MD, United States of America, 2 Centre for the AIDS Programme of Research in South Africa (CAPRISA), Nelson R Mandela School of Medicine, University of KwaZulu-Natal, Durban, South Africa, 3 South Africa Medical Research Council (SAMRC)-CAPRISA HIV-TB Pathogenesis and Treatment Research Unit, Durban, South Africa, 4 Centre for the AIDS Programme of Research in South Africa (CAPRISA), Umlazi Clinical Research Site, Durban, South Africa, 5 University of Zimbabwe College of Health Sciences Clinical Trials Research Centre, Harare, Zimbabwe, 6 Perinatal HIV Research Unit, University of the Witwatersrand, Johannesburg, South Africa, 7 University of North Carolina (UNC) Project-Malawi, Lilongwe, Malawi, 8 UNC-CH Department of Obstetrics and Gynecology, Chapel Hill, NC, United States of America, 9 Makerere University-Johns Hopkins University (MUJHU) Research Collaboration, Kampala, Uganda, 10 Department of Pathology, Johns Hopkins School of Medicine, Baltimore, MD, United States of America

¶ The complete membership of the author group can be found in the Acknowledgments.
* jaizire1@jhu.edu

**Data Availability Statement:** We uploaded an anonymized minimally sufficient data file to

## Abstract

### Background

About 90% of unintended pregnancies are attributed to non-use of effective contraception–tubal ligation, or reversible effective contraception (REC) including injectables, oral pills, intra-uterine contraceptive device (IUCD), and implant. We assessed the prevalence of unintended pregnancy and factors associated with using RECs, and Long-Acting-Reversible-Contraceptives (LARCs)–implants and IUCDs, among women living with HIV (WLHIV) receiving antiretroviral therapy (ART).

### Methods

We conducted cross-sectional analyses of the US-PEPFAR PROMOTE study WLHIV on ART at enrollment. Separate outcome (REC and LARC) modified-Poisson regression models were used to estimate prevalence risk ratio (PRR) and corresponding 95% confidence interval (CI).

### Results

Of 1,987 enrolled WLHIV, 990 (49.8%) reported their last/current pregnancy was unintended; 1,027/1,254 (81.9%) non-pregnant women with a potential to become pregnant reported current use of effective contraception including 215/1,254 (17.1%) LARC users.

replicate our study finding which can be accessed at DOI 10.6084/m9.figshare.25215209.

**Funding:** The PROMOTE study is funded by the President's Emergency Plan for AIDS Relief (PEPFAR) through DAIDS/NIAID/NIH grants to each of the following Clinical Trials Units (CTUs): JHU-Uganda CTU Makerere University-Johns Hopkins University (MU-JHU) Research Collaboration, grant # UM1 AI069530-11; The Johns Hopkins University-Blantyre Clinical Trials Unit, grant # UM1AI069518-12; The University of North Carolina Global HIV Prevention and Treatment Clinical Trials Unit, grant # 5UM1AI069423-12; University of Zimbabwe College of Health Sciences Clinical Trials Research Centre, grant # 5UM1AI069436-12; PHRU KARABELO Clinical Trials Unit for NIAID Networks Grant # 5UM1AI069453; Clinical Trials Unit for AIDS/Tuberculosis Prevention and Treatment - Grant Number: 5 UM1AI069469-11; and CAPRISA Clinical Trials Unit for AIDS/Tuberculosis Prevention and Treatment, grant # 5UM1 AI069469.

**Competing interests:** The authors declare that they have no competing interests including financial, consultant, institutional or other relationships that might lead to a bias or conflict of interest. This work was presented in part at the 10th International Workshop on HIV Pediatrics (abstract #83), 20-21 July 2018, Amsterdam, The Netherlands.

Compared to Zimbabwe, REC rates were similar in South Africa, aPRR = 0.97 (95% CI: 0.90–1.04), p = 0.355, lower in Malawi, aPRR = 0.84 (95% CI: 0.78–0.91), p<0.001, and Uganda, 0.82 (95% CI: 0.73–0.91), p<0.001. Additionally, REC use was independently associated with education attained, primary versus higher education, aPRR = 1.10 (95% CI: 1.02–1.18), p = 0.013; marriage/stable union, aPRR = 1.10 (95% CI: 1.01–1.21), p = 0.039; no desire for another child, PRR = 1.10 (95% CI: 1.02–1.16), p = 0.016; infrequent sex (none in the last 3 months), aPRR = 1.24 (95% CI: 1.15–1.33), p<0001; and controlled HIV load ($\leq$ 1000 copies/ml), PRR = 1.10 (95% CI: 1.02–1.19), p = 0.014. LARC use was independently associated with country (Zimbabwe ref: South Africa, PRR = 0.39 (95% CI: 0.26–0.57), p<0.001; Uganda, PRR = 0.65 (95% CI: 0.42–1.01), p = 0.054; and Malawi, aPRR = 0.87 (95% CI: 0.64–1.19), p = 0.386; HIV load ($\leq$ 1000 copies/ml copies/ml), aPRR=1.73 (95% CI: 1.26–2.37), p<0.001; and formal/self-employment, aPRR = 1.37 (95% CI: 1.02-1.91), p = 0.027.

## Conclusions

Unintended pregnancy was common while use of effective contraception methods particularly LARCs was low among these African WLHIV. HIV viral load, education, sexual-activity, fertility desires, and economic independence are pertinent individual-level factors integral to the multi-level barriers to utilization of effective contraception among African WLHIV. National programs should prioritize strategies for effective integration of HIV and reproductive health care in the respective African countries.

## Introduction

Sub-Saharan Africa is disproportionately affected by HIV and sexual and reproductive health (SRH) related morbidity/mortality burdens, accounting for more than 90% of the antiretroviral therapy (ART) and SRH needs globally [1, 2]. About 60% of adults living with HIV are women of reproductive age, and reports from several countries in the region suggest that about half of all pregnancies are unintended [1–3]. This situation was potentially aggravated by the COVID-19 pandemic travel restrictions and supply disruptions [4]. The majority (90%) of unintended pregnancies in the region among women with contraceptive need are attributed to non-use of effective contraception [5]. The World Health Organization classifies contraception methods associated with less than 1% failure (unintended pregnancy) rate in the first year as very effective. These include tubal ligation, vasectomy, implants, IUCD, injectables, and methods that require consistent and correct user action such as oral pills and six-month lactational amenorrhea. The second-tier methods associated with 1–9% failure rates are classified as effective and include consistent and correct use of condoms and calendar methods [6]. Both the IUCD and implant classified as long-acting reversible contraceptives (LARCs) are the most effective reversible contraception for prevention of unintended pregnancy and the associated abortion risks and short pregnancy intervals [6–8].

While unmet contraception need is defined as non-use of any contraception among women with a desire to avoid pregnancy [9], about six-in-ten women in the region have an unmet need for effective contraception. Traditional methods such as periodic abstinence, withdrawal, breastfeeding or douching which remain relatively common in these settings, are unreliable and are associated with high contraceptive failure rates [10]. Common reasons for

not using effective contraception include infrequent-sex, safety/side-effects/inconvenience, postpartum amenorrhea/breastfeeding, and partner or access related reasons [11].

Among women living with HIV (WLHIV) in Africa, reproductive health decision-making and attitudes towards pregnancy-intent are potentially further complicated by unique and dynamic factors. For example, ART initiation in women with very low CD4 cell counts was associated with changes in fertility desires upon realization of improved chances of delivering an HIV-free baby, resumption of sexual libido with improving health, and social-economic and partner-related expectations [12–15].

Unfortunately, a high frequency of rapid repeat pregnancies would potentially undermine the overall goal of ART programs to maintain maternal and child health. For example, in the context of highly effective ART programs, the more babies that are born to WLHIV will equate to higher absolute number of vertical transmissions. Unintended pregnancies are associated with adverse obstetric outcomes such as pre-term delivery and low birth weight, and maternal complications including maternal depression and maternal depletion syndrome–a state of poor maternal and fetal/child health attributed to nutritional stress as a result of rapid repeat pregnancy with short pregnancy intervals and limited recovery times [16–18]. This article describes the reported intendedness of last pregnancy and factors associated with using effective contraceptives in a multi-site cohort of African WLHIV on life-long ART in four of the highest burden countries in sub-Saharan Africa.

## Methods

### Study population

This was a cross-sectional analysis of the PROMOTE *(PROMise Ongoing Treatment Evaluation)* baseline (study entry) data. The PROMOTE study was a five-year follow-up of WLHIV on lifelong ART to assess long-term maternal health and the study design and procedures were previously described [19]. Eligible participants were enrolled between December 2016 and June 2017 at eight research sites in Malawi, Uganda, South Africa, and Zimbabwe. All pregnant and non-pregnant WLHIV who were in follow-up in the PROMISE (Promoting maternal and Infant Survival Everywhere) trial at the respective sites [20, 21], were transitioned and recruited/continued follow up in the PROMOTE study. In summary, PROMOTE study eligibility criteria included residence stability with no plans to migrate during the study follow-up period; and written informed consent for mother and/or child provided by the mother/caregiver. Both local and US affiliated Institutional Review Boards provided ethical research oversight. The study was funded by the United States Presidential Emergency Program for AIDS Relief (US-PEPFAR).

**PROMOTE study procedures.** Standardized questionnaires were used to collect sociodemographic, reproductive-health, and clinical data. All women were screened for pregnancy based on self-reported last normal menstrual period dates, obstetric examination and/or laboratory detection of human chorionic gonadotropin (hCG) hormone in the urine per standard-of-care guidance and locally available commercial urine hCG test kits. Trained study staff provided ongoing ART and effective contraception counseling and linkage-to-care (outside the study) per country-specific guidelines.

### Statistical methods

This was a planned cross-sectional analysis of the baseline PROMOTE study data. The outcome of interest was effective contraceptive use analyzed separately as reversible effective contraceptive (REC) use, and Long-Acting-Reversible-Contraceptive (LARC) use. Contraceptive type was assessed by asking current users, *"What family planning method(s) was used? (Mark*

*all that apply)".* Documentary evidence from the standard-of-care provider was sought to verify reported contraception method, and a certified copy was maintained on participant file. For purposes of this analysis, women who self-reported implant, IUD, injectables, or oral pills were classified as REC users; and women who reported implant or IUD methods as LARC users. Potentially effective contraceptive methods including condoms, lactational amenorrhea or calendar methods were classified as non-RECs because we did not have data to verify consistent and correct use. Other reported methods classified as non-RECs included abstinence, withdrawal, and breastfeeding.

The following characteristics which were determined *aprior* based on background knowledge and literature review were measured and analyzed to assess associations with REC and LARC use respectively. Socio-demographic factors included country of residence, age, education, household electricity use (as a measure of socio-economic status), marital/stable sexual partnerships, and employment status; reproductive-health factors included unintended last pregnancy, recent sexual activity, future fertility desire (*"Would you like to have another child?"*), and travel time to access reproductive health services; and HIV load. Women were classified as pregnant or non-pregnant, and if non-pregnant, classified by pregnancy potential i.e., women of reproductive age with no prior history of hysterectomy nor tubal ligation. Unintended last pregnancy was assessed by asking women, *"When you became pregnant this time (if currently pregnant) or last time you became pregnant, did you want to become pregnant then, or did you want to become pregnant later, or did you not want to become pregnant?"*. The last pregnancy was classified as unintended if a woman reported it was unintended, or wanted it delayed. To assess sexual activity, women were asked, *"In the last 3 months did you have sexual intercourse?"* and a "no" response was classified as infrequent sexual activity.

Overall, and country specific descriptive summary statistics were analyzed using proportions for categorical variables and median (inter-quartile range [IQR]) for continuous and discrete variables. Wilcoxon Rank Sum and Fisher's exact test were used to compare continuous or discrete and categorical variables, respectively, across REC or LARC groups. To estimate the measures of association at study enrollment, separate REC and LARC outcome multivariable modified Poisson regression models were used to measure point and 95% confidence interval (CI) prevalence risk ratio (PRR) estimates, and corresponding *p*-values based on two-sided hypothesis tests with type-1 error ($\alpha = 0.05$). Based on epidemiological rationale which were specified *a priori* we included all the above-mentioned variables in the multivariable regression models. Variables were analyzed as follows: country (ref: Zimbabwe versus Malawi, Uganda, South Africa); age group (ref: $\geq$30 years versus 20–24, 25–29); unintended last pregnancy (ref: no); desire for another child (ref: yes versus no, not sure); infrequent sexual activity (no sex in last 3 months versus yes); married/ regular partner (ref: single/ divorced/ widowed/ separated); primary education (ref: secondary and beyond); formal employment/self-employed (ref: stay-home / no formal employment); no household electricity use (ref: yes); clinic travel time $\geq$ 1 hour (ref: < 1 hour); and HIV load $\leq$1000 copies/ml (ref:>1000). Data were analyzed using SAS, version 9.4(SAS Institute Inc).

## Results

### Study profile and population characteristics

A total of 1,987 WLHIV were enrolled. Overall, 1,944 (97.8%) women reported currently taking ART with similarly high proportions reported across sites. The median (IQR) ART duration at PROMOTE study entry was 5 (4–5) years. The majority 1,462 (84.7%) reported currently taking efavirenz (EFV) based ART. A high (>90%) proportion of EFV-based ART was uniform across countries except for Uganda with 26.2% reporting EFV based ART, 51.0%

lopinavir/ritonavir-based ART, and 22.8% were using other regimens. Overall, 70 (3.5%) women had WHO class 3 or 4 HIV clinical disease. The median CD4 cell count was high, 825.5 cells/uL, which was statistically different across countries (Uganda: 963.5 cells/uL; Malawi: 739 cells/uL; Zimbabwe: 911.5 cells/uL; South Africa: 794 cells/uL; p<0.001). Viral load was not detectable in 1,628 (85%) of 1,680 women with viral load measures. Of the 301 women with detectable VL, 78 (25.9%) had VL less than 200 copies/ml, 21 (7.0%) had VL between 200–399 copies/ml, 31 (10.3%) had VL between 400–1000 copies/ml and 171 (56.8%) women had VL greater than 1000 copies/ml.

All the enrolled women had at least one prior pregnancy with a median (IQR) of 3 (2–3) surviving children. Of the enrolled women, 215 (10.8%) were currently pregnant or 6 weeks post-partum, 1,765 (88.8%) non-pregnant women had a biological potential (no prior hysterectomy nor tubal ligation) to become pregnant, and 7 (0.4%) non-pregnant women had a prior hysterectomy or tubal ligation (Fig 1). Non-pregnant women with a pregnancy potential and whose current contraception use data were available (n = 1,254) contributed to the contraception use analysis. Demographic, socio-economic, and reproductive health characteristics are summarized by REC use (Table 1). The median [IQR] age was 31 [28–35] years. Most (78.2%) women had completed primary education, about half (45.0%) were either unemployed or stay-at-home. Majority (87.8%) were married or in a regular partnership, 10% reported sexual activity in the previous 3 months, and about one-in-three desired another child. The

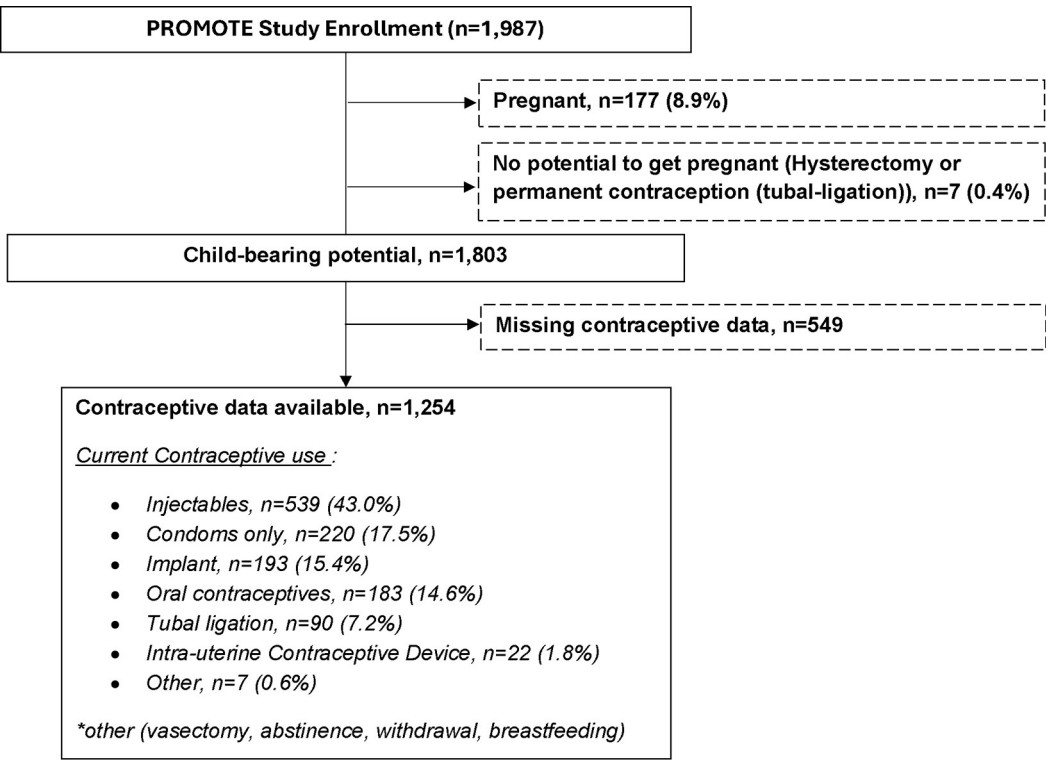

**Fig 1. PROMOTE study enrollment and contraceptive use reported by women with available data at baseline.**
PROMOTE—Promoting Maternal and Infant Survival Everywhere; Child-bearing potential refers to women who had no biological preclusion to becoming pregnant; Of the 1,254 women with available contraception data at baseline, every woman reported currently using at least one method to prevent getting pregnant. Contraceptive data was not available for 549 (30.4%) women who were enrolled before the relevant reproductive health questions were added to the baseline questionnaire.

**Table 1.  Study population characteristics at study enrolment among women with pregnancy potential, by reversible effective contraception (REC) use.**

| Characteristics | Overall | REC use | No REC use | P-value |
|---|---|---|---|---|
| | (N = 1,254) | (N = 1,027) | (N = 227) | |
| *Maternal characteristics* | | | | |
| Country, n (%) | | | | <0.001 |
| Uganda | 183 (14.6) | 132 (12.9) | 51 (22.5) | |
| Malawi | 425 (33.9) | 338 (32.9) | 87 (38.3) | |
| Zimbabwe | 302 (24.1) | 264 (25.7) | 38 (16.7) | |
| South Africa | 344 (27.4) | 293 (28.5) | 51 (22.5) | |
| Age (years), median [IQR] | 31 (28–35) | 31 (28–35) | 30 (26–35) | 0.095 |
| Age group (years), n(%) | | | | 0.114 |
| 20–24 | 122 (9.7) | 93 (9.1) | 29 (12.8) | |
| 25–29 | 360 (28.7) | 290 (28.2) | 70 (30.8) | |
| ≥30 | 772 (61.6) | 644 (62.7) | 128 (56.4) | |
| Highest level of education, n(%) | | | | 0.477 |
| None/Primary | 273 (21.8) | 228 (22.2) | 45 (19.8) | |
| Secondary and higher | 981(78.2) | 799 (77.8) | 182 (80.2) | |
| Employment, n (%) | | | | 0.995 |
| Formal employment | 292 (23.3) | 239 (23.3) | 53 (23.3) | |
| Self-employment (small business) | 396 (31.6) | 325 (31.7) | 71 (31.3) | |
| Not formally employed/housewife | 564 (45.0) | 461 (45.0) | 103 (45.4) | |
| Electricity in household, n (%) | 872 (69.5) | 707 (68.8) | 165 (72.7) | 0.266 |
| Married/regular partner, n (%) | 1101 (87.8) | 904 (88.0) | 197 (86.8) | 0.577 |
| Unintended last pregnancy, n (%) | 643 (51.4) | 544 (53.0) | 99 (43.8) | **0.013** |
| No sex in the last 3 months, n (%) | 127 (10.1) | 121 (11.8) | 6 (2.6) | <0.001 |
| Desire for another child, n (%) | | | | <0.001 |
| Yes | 461 (36.8) | 352 (34.3) | 109 (48.2) | |
| No | 711 (56.7) | 606 (59.0) | 105 (46.5) | |
| Not sure | 81 (6.5) | 69 (6.7) | 12 (5.3) | |
| Clinic travel time <1 hour, n (%) | 902 (72.0) | 751 (73.1) | 151 (66.8) | 0.06 |
| Number of surviving children, median (IQR) | 3 (2–3) | 3 (2–3) | 2 (2–4) | 0.100 |
| Number of surviving children, n(%) | | | | 0.211 |
| 0–2 | 594 (47.4) | 478 (46.6) | 116 (51.3) | |
| ≥3 | 658 (52.6) | 548 (53.4) | 110 (48.7) | |
| Currently on ART, n (%) | 1223 (97.5) | 1002 (97.6) | 221 (97.4) | 0.814 |
| Viral load ≤1000 copies/ml, n (%) | 1142 (91.1) | 929 (90.5) | 213 (94.2) | 0.071 |
| CD4 count (cells/mm$^3$), median (IQR) | 826 (648–1039) | 829 (649–1041) | 814.5 (643–1027) | **0.859** |

REC–Reversible Effective Contraceptives including oral, injectable, intra-uterine device (IUD), or implants; [5] IQR, Interquartile range; [a] *p*-value from Wilcoxon rank-sum test; [b] p-value from Fisher's exact test. Highlighted p-values are statistically significant. The number of participants with missing data was minimal: pregnancy-intent (n = 2); employment (n = 2); maternal-age (n = 1); fertility desire (n = 1); clinic travel-time (n = 1); surviving children (n = 2); CD4 count (n = 9) and no missing data on education; electricity-use; marital-status; sexual history; ART status; and viral load.

distributions of country (p<0.001), unintended last pregnancy (53.0% vs. 43.8%, p = 0.013), sexual activity (11.8% vs. 2.6%, p<0.001), and desire for another child (34.3% vs. 48.2%, p<0.001) were heterogenous across REC use categories. Age, education, employment, household electricity, marital status, clinic travel time, surviving children, ART initiation, and viral load were homogenously distributed by REC status. Similar distribution patterns were observed in the subset of women (n = 511 (29.0%)) who did not have reproductive-health data and were excluded from the correlates of contraceptive use analysis.

### Prevalence of unintended last pregnancy

At enrollment, 177/1,987 (8.9%) women were pregnant, of these 80 (45.2%) reported it was unintended. Overall, 990/1987 (49.8%) of the enrolled women reported their last pregnancy was unintended. Of these, 409 (41.3%) reported pregnancy was mistimed and 581 (58.7%) pregnancy was unwanted. There were site differences (p<0.001). The highest prevalence of unintended last pregnancy was observed at the South African sites (Durban [81.9%], and Soweto [55.7%]); followed by the Malawi sites (Lilongwe [57.5%], and Blantyre [51.3%]); the Zimbabwe sites ranged from 29.8% to 41.9%; and 28.4% at the Uganda site.

### Reported contraceptive use

Among 1,254 (63.1%) WLHIV who were not pregnant at enrolment and with a biological potential to become pregnant and with available contraceptive data), 1,027 (81.9%) self-reported REC use including 215 (17.1%) on LARCs. Injectables were the commonest 43.0%, followed by condoms only 17.5%, implant 15.4%, oral pills 14.6%, tubal ligation 7.1%, and IUD 1.8%. Of the 1,027 women, 583 (56.8%) reported dual use of condoms together with another REC. There were country variations (p<0.001) in REC use. Injectables were the commonest choice reported in Uganda (41.5%), Malawi (44.9%), and South Africa (60.2%), while oral pills the most reported in Zimbabwe (44.0%). A similar frequency pattern of injectables followed by condoms only, implant and oral pills, was reported in Malawi, Uganda, and South Africa. In Zimbabwe oral pills were the commonest followed by injectables, implants, and condoms only (Fig 2). Overall, traditional methods including abstinence, withdrawal, or breast-feeding, were the only contraception reported by about one-in-five women: highest in Uganda, 31.1% and Malawi, 24%, followed by South Africa, 17.4%, and 12.2% in Zimbabwe.

### Factors associated with contraceptive (REC and LARC) use

Separate (REC and LARC outcomes) multivariable model results are presented in Table 2. Compared to Zimbabwe, rates of REC use were similar in South Africa, aPRR = 0.97 (95% CI: 0.90–1.04), p = 0.355, and significantly lower in Malawi, aPRR = 0.84 (95% CI: 0.78–0.91), p<0.001) and Uganda, 0.82 (95% CI: 0.73–0.91), p<0.001. In addition, REC use was associated with *primary versus higher education, aPRR = 1.10 (95% CI: 1.02–1.18), p = 0.013;* marriage/ stable union, aPRR = 1.10 (95% CI: 1.01–1.21), p = 0.039; no desire for another child, aPRR = 1.10 (95% CI: 1.02–1.16), p = 0.016; infrequent sex (no sex in the last 3 months), aPRR = 1.24 (95% CI: 1.15–1.33), p<0001; as well as controlled HIV load below or equal to 1000 viral copies/ml, aPRR = 1.10 (95% CI: 1.02–1.19), p = 0.014.

Regarding use of LARCs, lower rates were observed in South Africa, Malawi, and Uganda relative to Zimbabwe; significantly lower in South Africa, aPRR = 0.39 (95% CI: 0.26–0.57), p<0.001, borderline significance in Uganda, aPRR = 0.65 (95% CI: 0.42–1.01), p<0.054, while the lower rates in Malawi were not statistically significant, aPRR = 0.87 (95% CI: 0.64–1.19), p = 0.386. Additionally, LARC use was associated with controlled HIV load (below or equal to 1000 viral copies/ml), aPRR = 1.73 (95% CI: 1.26–2.37), p<0.001; and *formal/self-employment versus stay home or no formal income, aPRR = 1.37 (95% CI: 1.02–1.91),* p = 0.027.

## Discussion

These cross-sectional data from a large multi-country cohort of African WLHIV receiving ART for life reveal a high prevalence (about half) of unintended last pregnancy and low rates of current use of effective contraception among those with contraception need. About one-in-five women with contraceptive need reported use of traditional methods which are known to be ineffective,

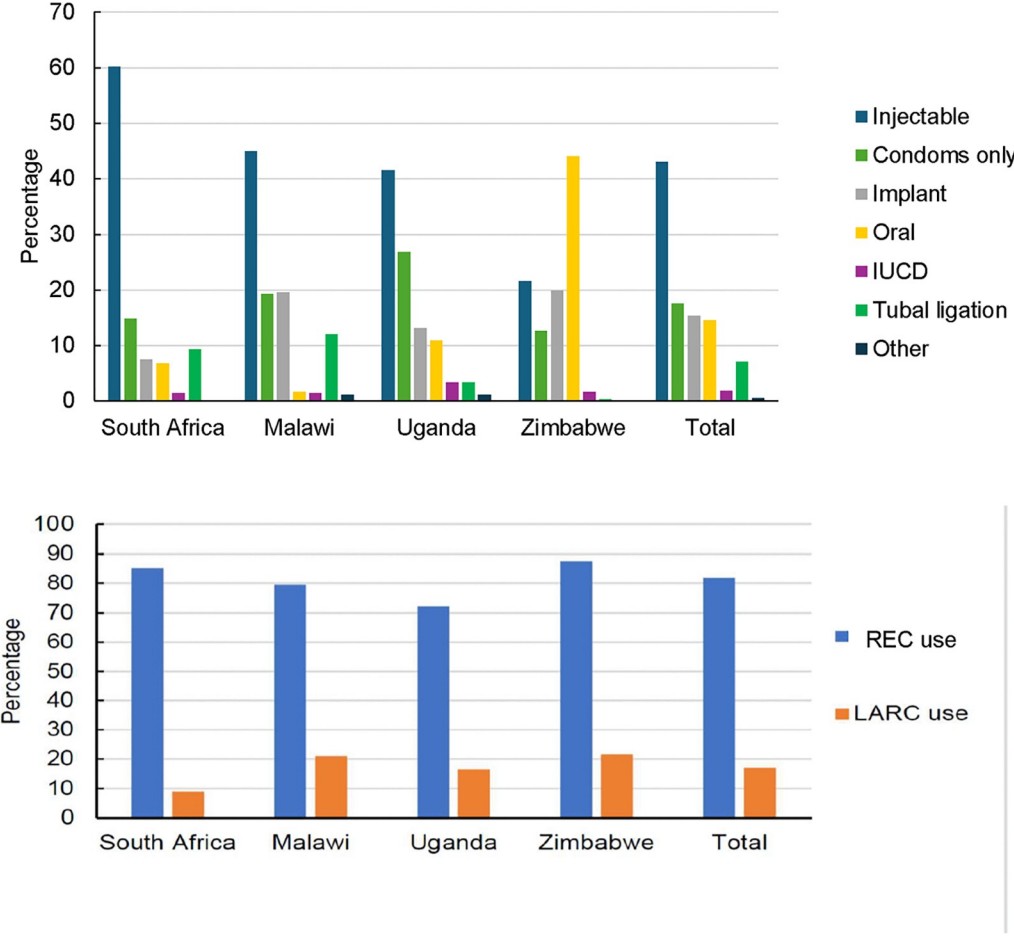

**Fig 2. Contraceptive methods reported by women using contraception and with available data at PROMOTE study entry.** Current contraceptive used reported by 1,254 women with available data at PROMOTE study entry. All these women reported currently using at least one method to prevent getting pregnant. REC–Reversible Effective Contraception including oral, injectable, intra-uterine contraceptive device (IUCD), or implants [5]; LARC—Long Acting Reversible Contraceptives including implant and IUCD which are considered to be the most effective reversible contraception for prevention of unintended pregnancy and the associated abortion risks and short intervals between pregnancies [7, 8]; *The category 'Other contraceptive methods' includes *abstinence, withdrawal, breastfeeding*.

and use of long-acting reversible contraceptives (LARCs) was very low (~20%). Based on multivariable regression analyses, marriage/stable relationship, sexual activity, having no desire for more children, relative HIV control (viral load ≤1000 HIV copies/ml), and primary versus higher education were independently associated with use of reversible effective contraceptives (RECs); but no associations were observed with age, socio-economic factors (employment, household electricity), unintended last pregnancy, and clinic travel-time. LARC use was associated with relative economic independence (formal/self-employment) and HIV-control, but not the other factors. Compared to Zimbabwe, rates of REC use were similar in South Africa but lower in Malawi and Uganda; and LARC rates were lower in South Africa, Malawi, and Uganda.

The unacceptably high unintended pregnancy rates observed in this study, which varied considerably across countries, are comparable to high prevalence (35–70%) reports from previous African WLHIV studies in similar resource-limited settings in Rwanda (2007), Botswana (2010–2012), South Africa (2009–2010), Eswatini (2010), Zimbabwe (2012), and Kenya (2016) [22–29]. However, the extremely high prevalence observed in this study at the Umlazi, Durban

**Table 2. Factors associated with use of contraceptive methods (REC/ LARC) reported at PROMOTE study entry.**

| | REC use | | LARC use | |
| --- | --- | --- | --- | --- |
| | **Adjusted PRR** **(95% CI)** | *P value* | **Adjusted PRR** **(95% CI)** | **P value** |
| Country (ref: Zimbabwe) | | | | |
| Malawi | **0.84 (0.78–0.91)** | **<0.001** | 0.87 (0.64–1.19) | 0.386 |
| South Africa | 0.97 (0.90–1.04) | 0.355 | **0.39 (0.26–0.57)** | **<0.001** |
| Uganda | **0.82 (0.73–0.91)** | **<0.001** | 0.65 (0.42–1.01) | 0.054 |
| Age group (years) (ref:≥30) | | | | |
| 20–24 | 0.95 (0.85–1.05) | 0.307 | 0.87 (0.54–1.39) | 0.552 |
| 25–29 | 1.00 (0.94–1.06) | 0.950 | 1.19 (0.91–1.56) | 0.211 |
| Unintended last pregnancy (ref: no) | 0.97 (0.91–1.02) | 0.192 | 0.89 (0.70–1.14) | 0.366 |
| Desire another child (ref: Yes) | | | | |
| No | **1.10 (1.02–1.16)** | **0.016** | 0.93 (0.71–1.23) | 0.680 |
| Unsure | 1.10 (0.98–1.21) | 0.097 | 0.84 (0.48–1.45) | 0.527 |
| Infrequent sex (self-reported no sex in last 3 months) | **1.24 (1.15–1.33)** | **<0.001** | 1.03 (0.69–1.53) | 0.861 |
| Married/ regular partner (ref: single/ divorced/widowed/separated) | **1.10 (1.01–1.21)** | **0.039** | 0.79 (0.56–1.10) | 0.159 |
| Primary or no education (ref: Secondary and beyond) | **1.10 (1.02–1.18)** | **0.013** | 1.23 (0.91–1.67) | 0.199 |
| Formal/self-employment (ref: Stay home women/ no formal employment | 1.04 (0.97–1.11) | 0.169 | **1.37 (1.02–1.91)** | **0.027** |
| No electricity in household | 1.04 (0.98–1.11) | 0.210 | 0.78 (0.58–1.04) | 0.090 |
| Clinic travel time ≥ 1 hour (ref: < 1 hour) | 0.98 (0.91–1.04) | 0.475 | 0.91 (0.69–1.21) | 0.525 |
| HIV load, ≤1000 copies/ml (ref:>1000 copies/ml) | **1.10 (1.02–1.19)** | **0.014** | **1.73 (1.26–2.37)** | **<0.001** |

REC–Reversible Effective Contraception including oral, injectable, intra-uterine contraceptive device (IUCD), or implants [5]; LARC—Long Acting Reversible Contraceptives including implant and IUD which are considered to be the most effective reversible contraception for prevention of unintended pregnancy and the associated abortion risks and short intervals between pregnancies [7, 8]; The respective multivariate regression models included all the variables listed: country (ref: Zimbabwe versus Malawi, Uganda, South Africa); age group (ref: ≥30 years versus 20–24, 25–29); unintended last pregnancy (ref: no); desire for another child (ref: yes versus no, not sure); infrequent sexual activity (no sex in last 3 months versus yes); married/ regular partner (ref: single/ divorced/ widowed/ separated); secondary education and beyond (ref: primary or no education); stay-home / no formal employment (ref: formal employment/self-employed); no household electricity use (ref: yes); clinic travel time ≥ 1 hour (ref: < 1 hour); and HIV load ≤1000 copies/ml (ref:>1000).

site in South Africa (81.9%) has not been reported previously. Moreover, it is likely that the prevalence observed in this urban/peri-urban study population is a conservative estimate of the respective country-level burdens since access to reproductive health services in many African countries is reportedly more than two-fold higher in urban versus rural settings where the majority of the population resides [30]. It has also been reported that WLHIV in these African settings who have not initiated lifelong ART or women who are not aware of their HIV infection status have a higher likelihood of reporting unintended pregnancy [23, 24, 29]. All the women in this study were aware of their HIV infection status and the majority (>97%) had initiated ART by the time they conceived the reported unintended last pregnancy. Data on contraceptive use prior to the reported unintended pregnancy was not available for this analysis which precluded assessment of other risk factors of unintended pregnancy among WLHIV reported in these settings including unmet need for effective contraceptives or contraceptive failures [25, 26, 29].

Regarding contraceptive use in this study population, traditional methods (breastfeeding, withdrawal, abstinence, calendar, or condoms only) which are potentially ineffective particularly when used incorrectly and/or inconsistently [6], were common (~20%), and reported LARC use was unacceptably low. These findings are concerning. Study participants transitioned from the PROMISE trial, a controlled five year follow-up study setting during which counselling and referral to standard-of-care reproductive-health clinics was routine [20, 21],

although it is worth noting that country-level and health facility barriers in these resource lim-ited settings might have restricted access/availability. Nonetheless, contemporaneous studies in these African settings reported comparable high rates of using potentially ineffective con-traceptive methods and similarly low LARC use [25, 28, 31–34]. The high prevalence of a single contraceptive method at the respective sites is consistent with reports from other African set-tings, compared to other regions which tend to report relatively more diverse contraceptive method-mix [25, 33, 35]. Since our study women received their contraceptives from standard-of-care/non-study providers, the observed skewed patterns may be an artifact of the policy, provider-factors (training, bias and attitudes) or available stocks at country, regional or facility level. Unfortunately, the predominantly reported methods in this study, both injectable- and oral-contraceptive methods are heavily user-action dependent, including 3-monthly provider-clinic visits for repeat injections, or require diligently swallowing daily oral contraceptives cou-pled with routine provider-clinic visits for refills. Intermittent non-adherence over a pro-tracted period of self-perceived contraceptive use may affect contraceptive effectiveness to low levels potentially comparable to non-contraceptive use [36]. Implants and IUCDs, which are the most effective reversible contraceptive methods for an extended period without requiring user-action were rarely reported: about one-in-ten of our study women reported implant use in South Africa and Uganda, and about one-in-five in Malawi and Zimbabwe; and less than 4% reported IUCDs. In these settings where provision of safe sterilization services is challeng-ing coupled with restrictive age eligibility criteria, LARCs are important alternatives for women of any age who particularly do not want to have more children. Unfortunately, implants and IUCDs require more provider skill and training. It has been suggested that women in these settings who are using short-acting hormonal methods might prefer a LARC if they had the opportunity [37], however, busy clinics may shun provision of LARCs since they are provider-labor-intensive. For example, the low enthusiasm towards implants by pro-viders in South Africa is attributed to a policy change that was deemed rushed with inadequate provider-training during the roll-out in 2014, followed shortly after by a policy against implant provision to women on efavirenz-based ART [38].

Collectively, our findings of high prevalence of unintended last pregnancy coupled with subsequent low REC usage rates, particularly LARCs, underscores the need to better under-stand individual-level, as well as community- and facility-level factors that may influence utili-zation of effective contraception by WLHIV in these settings. Concerningly, unintended last pregnancy did not appear to incentivize subsequent use of RECs, an indicator of perennial contextual factors that preclude utilization. Understandably, and consistent with previous reports, factors that were independently associated with REC use included having no desire for more children; [26, 32] being in a stable/sustained sexual partnership, and reported infre-quent sex independent of being in stable sexual partnership, both correlates of self-perceived risk of becoming pregnant; [32, 39] as well as relative HIV control [26], a proxy for good health. Indeed, WLHIV with relative HIV control adhering to HIV care and ART protocols are more likely to benefit from available reproductive health services such as RECs including LARCs. However, the negative association between REC use and higher education in this study is counterintuitive, similar to previous African studies linking higher education with unmet contraceptive need [40–42], and another suggesting education had no influence on effective contraceptive choice [43]. A plausible explanation is that WLHIV in our study with higher education delayed childbearing since higher education was correlated with desire for another child. The considerable variations in the contraception options reported, including REC as well as LARC use, across different countries in our study, is characteristic in the region. This heterogeneity reflects the existing substantial contextual dissimilarities [44], including level of political will and governments' investment and commitment to contraceptive services

across African countries [45]. It is not surprising that age was not associated with REC or LARC use in this study considering the relatively homogenous age group with a median (IQR) 31 (28–35) years of age. This is consistent with previous studies with similar age band comparisons [26, 40, 46]. Studies that reported associations between age and contraceptive use included younger women and adolescents, a group associated with higher unmet contraception needs [32, 40–42]. However, these studies did not specify use of effective contraception, and while they were conducted in high HIV-burden settings, individual HIV status was not ascertained. An Ethiopian study of WLHIV reported lower use of modern contraceptives among older women above 35 years compared to their 15–24 year old counterparts [46].

This study had some limitations. A recall bias inherent to cross-sectional surveys was likely in these analyses based on individuals thinking back and reporting their pregnancy intent, although the magnitude was likely minimal since as has been argued, pregnancy related events are relatively easier to recall given their significance in a woman's lifetime [47]. A woman's desire for a given pregnancy is fluid and intervening factors before, during and after pregnancy such as partner and/or other social-economic support, access to care, loss of the pregnancy or demise of an older child, etc., will likely influence the woman's attitude towards the pregnancy [48–51]. A tendency towards "retrospective rationalization" of previously undesired pregnancies has been proposed [49, 52] suggestive of underreporting of unintended pregnancy rates; although the reverse has been reported where women changed from intended to unintended pregnancy reports, in a repeated measure analysis [53]. Also, the lack of data on contraceptive-choices immediately preceding the reported unintended pregnancies precluded the ascertainment of whether the reported unintended pregnancies resulted from non-use of effective contraceptives or contraceptive failure including user-related (inconsistent or incorrect use) or method-related failures. In addition, since our study women received their contraceptives from standard-of-care clinics, and the data was not available for this analysis, we were not able to assess health facility, provider, or policy barriers to REC or LARC access. Also, study contraceptive data was not available for about 30.4% of the women enrolled before the relevant reproductive health questions were added to the baseline questionnaire. The exclusion of these women from these analyses may have potentially resulted in a selection bias. Reassuringly, the distribution of baseline characteristics was similar among the excluded women compared to those who contributed to these analyses. Additionally, the age range was limited. Exclusion of younger WLHIV who may have much higher rates of unintended pregnancy, and older WLHIV who may have lower rates of unintended pregnancy is a potential threat to the internal validity (selection bias) and external validity (generalizability) of these findings. It is worth noting that health system factors ranging from policy through primary health care delivery may influence access to effective contraceptive choices. Availability factors such as range of options, stock-outs, etc., were beyond the scope of this analysis. Contraception methods reported by WLHIV in this study were likely influenced by available options which may not be a true reflection of their preferred choice.

Nonetheless, these study findings contribute to the growing body of empirical data suggestive of reproductive health challenges that persist in resource-limited African settings, which also have a heavy HIV burden. This analysis was based on a large sample of WLHIV on life-long ART from very high burden countries. Data was well characterized including laboratory confirmation of HIV status, a challenge in previous studies which used simpler self-report data [25].

## Conclusions

Unacceptably high rates of unintended pregnancy persist among WLHIV on ART; concerningly, effective contraceptive methods were not universally reported among women with

contraceptive need. LARCS were rarely reported. There were considerable variations across the four eastern and southern African countries, an indication of cultural, social, and political contextual differences influencing access and utilization of RECs including LARCs. Unfortunately, the persistence of reproductive health challenges does not match recent considerable improvements in access and utilization of HIV care services in the respective countries. Country level efforts should employ integrated HIV care and reproductive health services delivery [54], while carefully considering the risk categorization: future fertility desires, sexual activity, self-perceived risk of unintended pregnancy, and socio-economic and cultural perspectives.

While it remains unlikely that women in many African settings will be availed the whole spectrum of effective contraception, it is imperative that national programs provide options for each of the main categories of contraception that are safe, user-friendly, and discrete, to WLHIV clients attending HIV care programs, including all care during pregnancy through delivery. This is essential to uphold reproductive health rights, and to mitigate health burden due to unintended pregnancy, unsafe abortion, and lost economic productivity of women which ultimately impacts family health [55–57]. LARCs, including immediate provision of LARCs to postpartum mothers [58], is a strategy that minimizes contraceptive adherence challenges attributed to user-action dependency. Research efforts to better understand potential dual protective benefits of multipurpose technologies–involving adding ART and a contraceptive agent such as levonorgestrel using barrier devices such as intravaginal rings, should be prioritized [59].

## Acknowledgments

The authors acknowledge and thank all the study participants in Malawi, Uganda, South Africa and Zimbabwe, and the dedicated conduct of the study by the US-PEPFAR PROMOTE study team at the respective sites, under the overall leadership of co-chairs Prof. Taha Taha and Prof. Mary Glenn Fowler. We acknowledge the teams at each of the following research sites and coordination centers: Centre for the AIDS Programme of Research in South Africa (CAPRISA), Nelson R Mandela School of Medicine, University of KwaZulu-Natal, Durban, South Africa; South Africa Medical Research Council (SAMRC)-CAPRISA HIV-TB Pathogenesis and Treatment Research Unit, Durban, South Africa; Centre for the AIDS Programme of Research in South Africa (CAPRISA), Umlazi Clinical Research Site, Durban, South Africa; University of Zimbabwe College of Health Sciences Clinical Trials Research Centre, Harare, Zimbabwe; Perinatal HIV Research Unit, University of the Witwatersrand, Johannesburg, South Africa; University of North Carolina (UNC) Project-Malawi, Lilongwe, Malawi; UNC-CH Department of Obstetrics and Gynecology, Chapel Hill, NC, USA; Makerere University-Johns Hopkins University (MUJHU) Research Collaboration, Kampala, Uganda; Johns Hopkins School of Medicine, Department of Pathology, Baltimore, MD, USA; and Johns Hopkins Bloomberg School of Public Health, Department of Epidemiology Baltimore, MD, USA.

## Author Contributions

**Conceptualization:** Jim Aizire, Mary Glenn Fowler, Taha Taha.

**Data curation:** Jim Aizire, Nonhlanhla Yende-Zuma, Mary Glenn Fowler, Taha Taha.

**Formal analysis:** Jim Aizire, Nonhlanhla Yende-Zuma, Mary Glenn Fowler, Taha Taha.

**Funding acquisition:** Mary Glenn Fowler, Taha Taha.

**Investigation:** Jim Aizire, Sherika Hanley, Teacler Nematadzira, Mandisa M. Nyati, Sufia Dadabhai, Lameck Chinula, Catherine Nakaye, Mary Glenn Fowler, Taha Taha.

**Methodology:** Jim Aizire, Mary Glenn Fowler, Taha Taha.

**Project administration:** Jim Aizire, Mary Glenn Fowler, Taha Taha.

**Supervision:** Jim Aizire, Sherika Hanley, Teacler Nematadzira, Mandisa M. Nyati, Sufia Dadabhai, Lameck Chinula, Catherine Nakaye, Mary Glenn Fowler, Taha Taha.

**Writing – original draft:** Jim Aizire.

**Writing – review & editing:** Jim Aizire, Sherika Hanley, Teacler Nematadzira, Mandisa M. Nyati, Sufia Dadabhai, Lameck Chinula, Catherine Nakaye, Mary Glenn Fowler, Taha Taha.

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
