## [Decision Letter · Decision Letter 0]

19 Apr 2023

PONE-D-23-00005Unintended Pregnancy and Contraception Use among African women living with HIV: baseline analysis of the multi-country US PEPFAR PROMOTE cohortPLOS ONE

Dear Dr. Aizire,

Thank you for submitting your manuscript to PLOS ONE. After careful consideration, we feel that it has merit but does not fully meet PLOS ONE’s publication criteria as it currently stands. Therefore, we invite you to submit a revised version of the manuscript that addresses the points raised during the review process.

In addition to reviewer #1 comments, please address the following comments:  Academic Editor Comments:

Please carefully re-read your submission and check for grammatical errors as there are several throughout including sentences that do not make sense/seem to be missing punctuation splitting them into multiple sentences.

Please double check IUCD vs IUD and use the more acronym. Also spell out at first use (in introduction).

Include eligibility criteria

Define criteria to be included in "biological potential to become pregnant"

How did you determine that mistimed pregnancies were unwanted, that doesn't seem to necessarily relate. If someone was trying to conceive, but it occured earlier or later than planned, it doesn't seem that it was an unwanted pregnancy.

Please rephrase unacceptably high as that implies judgment (page 12).

Not all women have access to more reliable forms of contraceptives, whether because of availability in country, cost, consent requirements, etc. Please remove unacceptably low language on page 12. It's not realistic to assume that all women in these countries have access to their desired or a more reliable form of birth control. Even if they do, gender roles can directly influence access and women may not have the power to use their preferred form of birth control.

We look forward to receiving your revised manuscript.

Kind regards,

Amee Schwitters, PhD

Academic Editor

PLOS ONE

Journal Requirements:

2.One of the noted authors is a group or consortium "US-PEPFAR PROMOTE Cohort Study team". In addition to naming the author group, please list the individual authors and affiliations within this group in the acknowledgments section of your manuscript. Please also indicate clearly a lead author for this group along with a contact email address.

Reviewers' comments:

Reviewer's Responses to Questions

**Comments to the Author**

1. Is the manuscript technically sound, and do the data support the conclusions?

Reviewer #1: Yes

2. Has the statistical analysis been performed appropriately and rigorously? 

Reviewer #1: Yes

3. Have the authors made all data underlying the findings in their manuscript fully available?

Reviewer #1: Yes

4. Is the manuscript presented in an intelligible fashion and written in standard English?

Reviewer #1: Yes

5. Review Comments to the Author

Reviewer #1: Abstract –

• Please separate the methods and results sections

• Methods: Cohort “at” enrolment?

• It’s not clear in the abstract if the Poisson regression was conducted with the enrolment cohort, or later after an intervention with this cohort. Perhaps clarify that you used it on the enrolment data only, and what period of recall was included (if relevant)

• Your methods need to reflect that enrolment was of women with >5 years on ART

• Findings: Close brackets after p=0.054. (and use ‘=’ and not ‘<’ since you’re being precise with the 0.054)

• Conclusions: the final sentence is grammatically incorrect – please separate the final segment of the sentence into its own sentence – looks like it’s meant to be a recommendation statement.

Introduction –

• “S&RH” or SRH? Please confirm norms for acronym in the literature

• Half of WLHIV pregnancies or half of all pregnancies are unintended?

• Read carefully for grammatical correctness.

• In the paragraph starting with ‘unfortunately’, I think the language needs to be revised. The clear implication as it’s currently stated is that we should reduce the number of babies that WLHIV should have – this is very politically incorrect. I suggest restructuring the sentence so the second reason comes first (maternal health), and then reframe it that even in a context of highly effective PMTCT, the more babies that are born to WLHIV will equate to a higher absolute number of vertical transmissions.

Methods –

• Study procedures – you need to mention the >5 years on ART. This is mentioned at the end of the intro, but should be clear here that it was an eligibility criterium

• Analysis – correct ‘a prior’ to ‘a priori’

• (as in the abstract comments) It’s not clear if the Poisson regression was conducted with the enrolment cohort, or later after an intervention with this cohort. Perhaps clarify that you used it on the enrolment data only, and what period of recall was included (if relevant). As most readers will be familiar with Poisson regression only being used for analysis over time, this could be confusing as it’s currently written.

Results -

• You present median duration as 5 years, but the final introductory sentence says “on ART for more than 5 years” and your methods doesn’t indicate anything about ART duration as enrolment criteria. Suggest you clarify across all sections.

• Where describing VL, please add percentages

• How do you define “pregnancy potential” and “a biological potential to be come pregnant”? Usually would be ‘of reproductive age’, but your language choice implies additional criteria – please ensure that definition is included in your methods if so. And if not, consider changing wording

• Marriage/Sex/wanting children – please remove the words “while” and “only” – this implies the addition of your personal opinion, rather than purely reporting the findings. Note – this is a very interesting finding, and perhaps worth a mention in the discussion. Quantitative findings give the “what” and it would be helpful if qualitative research could assist with answering the “why” in future studies.

Discussion -

• P15 – it’s not clear *why* your cohort has such a small age range (28-35) – if this is a function of them being a continuing cohort from a previous study, it would be helpful to describe that in the methods, and as a limitation.

• As a limitation – the limited age range leaves out the younger ages who may have much higher rates of unintended pregnancy, and the older ages, who may have lower rates of unintended pregnancy. Therefore your study has selection bias and can’t be generalized to all women of reproductive age in these countries.

Conclusion –

• Consider how the comments on choice/options could be interpreted by those in severely resource-constrained governments, and what message you’re trying to leave with the reader – in most of these settings, it’s not a realistic expectation that all women “are availed the whole spectrum of effective contraception options” – national health systems can’t afford this. But they should provide options for each of the main categories of contraceptive methods at least at the district hospital level (since it won’t be possible to always decentralize further).

• Variation in choice may also be influenced a lot by the health system – you can only access what’s been ordered and what’s in stock, whether it’s your preferred method or not.

6. PLOS authors have the option to publish the peer review history of their article (what does this mean?). If published, this will include your full peer review and any attached files.

Reviewer #1: **Yes: **Beth A Tippett Barr

---

## [Author Response · Author response to Decision Letter 0]

2 Jun 2023

On behalf of myself and my colleagues, we submit the revised version of the manuscript addressing your review comments. Thanks for your guidance in this regard. 

We made revisions throughout the manuscript to clarify the study population, statistical methods and expounded on some of the limitations, as well as grammatical errors. 

Below is a point-by-point response to the reviewers’ comments. Also attached is the revised manuscripts (tracked changes and clean versions). 

Review: Unintended pregnancy and contraception use among African women living with HIV: baseline analysis of the multi-country US PEPFAR PROMOTE cohort.

Abstract – 

• Please separate the methods and results sections

Response: Noted, thanks. 

• Methods: Cohort “at” enrolment?

Response: We revised the sentence to make this clearer. 

• It’s not clear in the abstract if the Poisson regression was conducted with the enrolment cohort, or later after an intervention with this cohort. Perhaps clarify that you used it on the enrolment data only, and what period of recall was included (if relevant)

Response: We made changes to clarify that analyses were based on enrollment data. 

• Your methods need to reflect that enrolment was of women with >5 years on ART

Response: We made pertinent revisions to reflect the study population of WLHIV on life-long ART. 

• Findings: Close brackets after p=0.054. (and use ‘=’ and not ‘<’ since you’re being precise with the 0.054)

Response: Noted, thanks. 

• Conclusions: the final sentence is grammatically incorrect – please separate the final segment of the sentence into its own sentence – looks like it’s meant to be a recommendation statement.

Response: We made revisions as suggested. The last sentence reads, ‘’ National programs should prioritize strategies for effective integration of HIV and reproductive health care in the respective African countries.’

Introduction – 

• “S&RH” or SRH? Please confirm norms for acronym in the literature

Response: We made revisions from ‘S&RH’ to ‘SRH’. 

• Half of WLHIV pregnancies or half of all pregnancies are unintended?

Response: We clarified the sentence as follows: “About 60% of adults living with HIV are women of reproductive age, and reports from several countries in the region suggest that about half of all pregnancies are unintended.”

• Read carefully for grammatical correctness.

Response: Noted, thanks. We made a few changes to improve the grammar. 

• In the paragraph starting with ‘unfortunately’, I think the language needs to be revised. The clear implication as it’s currently stated is that we should reduce the number of babies that WLHIV should have – this is very politically incorrect. I suggest restructuring the sentence so the second reason comes first (maternal health), and then reframe it that even in a context of highly effective PMTCT, the more babies that are born to WLHIV will equate to a higher absolute number of vertical transmissions.

Response: 

“Unfortunately, a high frequency of rapid repeat pregnancies would potentially undermine the overall goal of ART programs to maintain maternal and child health. For example, in the context of highly effective ART programs, the more babies that are born to WLHIV will equate to higher absolute number of vertical transmissions.”

Methods – 

• Study procedures – you need to mention the >5 years on ART. This is mentioned at the end of the intro, but should be clear here that it was an eligibility criterium. 

Response: We made revisions throughout the manuscript to clarify that the PEPFAR-PROMOTE study population were WLHIV on ART. 

The results section further clarifies the distribution of ART duration as follows: “The median (IQR) ART duration at PROMOTE study entry was 5 (4-5) years.”

• Analysis – correct ‘a prior’ to ‘a priori’

Response: Noted, thanks. 

• (as in the abstract comments) It’s not clear if the Poisson regression was conducted with the enrolment cohort, or later after an intervention with this cohort. Perhaps clarify that you used it on the enrolment data only, and what period of recall was included (if relevant). As most readers will be familiar with Poisson regression only being used for analysis over time, this could be confusing as it’s currently written.

Response: Noted, thanks. We made pertinent revisions, and the sentence reads: “To estimate the measures of association at study enrollment, separate REC and LARC outcome multivariable modified Poisson regression models were used to measure point and 95% confidence interval (CI) prevalence risk ratio (PRR) estimates, and corresponding p-values based on two-sided hypothesis tests with type-1 error (α = 0.05).”

Results -

• You present median duration as 5 years, but the final introductory sentence says “on ART for more than 5 years” and your methods doesn’t indicate anything about ART duration as enrolment criteria. Suggest you clarify across all sections.

Response: Thank you very much for this observation. WLHIV on PROMISE were receiving life-long ART, and ART use was not a criteria for transitioning to the PROMOTE study. We clarified the last sentence of the introduction as follows:

“This article describes the reported intendedness of last pregnancy and factors associated with using effective contraceptives in a multi-site cohort of African WLHIV with more than 5 years on life-long ART in four of the highest burden countries in sub-Saharan Africa.”

• Where describing VL, please add percentages.

Response: Thank you. We revised the sentence as follows: 

“Viral load was not detectable in 1,628 (85%) of 1,680 women with viral load measures. Of the 301 women with detectable VL, 78 (25.9%) had VL less than 200 copies/ml, 21 (7.0%) had VL between 200-399 copies/ml, 31 (10.3%) had VL between 400-1000 copies/ml and 171 (56.8%) women had VL greater than 1000 copies/ml.”

• How do you define “pregnancy potential” and “a biological potential to be come pregnant”? Usually would be ‘of reproductive age’, but your language choice implies additional criteria – please ensure that definition is included in your methods if so. And if not, consider changing wording.

Response: We clarified the definition of ‘non-pregnant women on page as follows:

“Women were classified as pregnant or non-pregnant, and if non-pregnant, classified by pregnancy potential i.e., women of reproductive age with no prior history of hysterectomy nor tubal ligation.”

• Marriage/Sex/wanting children – please remove the words “while” and “only” – this implies the addition of your personal opinion, rather than purely reporting the findings. Note – this is a very interesting finding, and perhaps worth a mention in the discussion. Quantitative findings give the “what” and it would be helpful if qualitative research could assist with answering the “why” in future studies.

Response:

On page 10: “Of the enrolled women, 215 (10.8%) were currently pregnant or 6 weeks post-partum, 1,765 (88.8%) non-pregnant women had a biological potential (no prior hysterectomy nor tubal ligation) to become pregnant, while and 7 (0.4%) non-pregnant women had a prior hysterectomy or tubal ligation.” 

On page 10: “While mMajority (87.8%) were married or in a regular partnership, only 10% reported sexual activity in the previous 3 months, and about one-in-three desired another child.”

On page 14: “Implants and IUCDs, which are the most effective reversible contraceptive methods for an extended period without requiring user-action were rarely reported: only about one-in-ten of our study women reported implant use in South Africa and Uganda, and about one-in-five in Malawi and Zimbabwe;

Discussion – 

• P15 – it’s not clear *why* your cohort has such a small age range (28-35) – if this is a function of them being a continuing cohort from a previous study, it would be helpful to describe that in the methods, and as a limitation. 

• As a limitation – the limited age range leaves out the younger ages who may have much higher rates of unintended pregnancy, and the older ages, who may have lower rates of unintended pregnancy. Therefore, your study has selection bias and can’t be generalized to all women of reproductive age in these countries. 

Response: Thank you very much for this observation. We made pertinent changes as follows: 

On page 7: “All pregnant and non-pregnant WLHIV who were in follow-up in the PROMISE (Promoting maternal and Infant Survival Everywhere) trial at the respective sites, [20], [21] were transitioned and recruited/continued follow up in the PROMOTE study.”

On page 17: “Additionally, the age range was limited. Exclusion of younger WLHIV who may have much higher rates of unintended pregnancy, and older WLHIV who may have lower rates of unintended pregnancy is a potential threat to the internal validity (selection bias) and external validity (generalizability) of these findings.”

Conclusion –

• Consider how the comments on choice/options could be interpreted by those in severely resource-constrained governments, and what message you’re trying to leave with the reader – in most of these settings, it’s not a realistic expectation that all women “are availed the whole spectrum of effective contraception options” – national health systems can’t afford this. But they should provide options for each of the main categories of contraceptive methods at least at the district hospital level (since it won’t be possible to always decentralize further). 

Response: Thank you very much for these observations. We made the suggested changes on page 18 as follows:

“While it remains unlikely that women in many African settings will be availed the whole spectrum of effective contraception, it is imperative that national programs provide options for each of the main categories of contraception that are safe, user-friendly, and discrete, to WLHIV clients attending HIV care programs, including all care during pregnancy through delivery.”

• Variation in choice may also be influenced a lot by the health system – you can only access what’s been ordered and what’s in stock, whether it’s your preferred method or not.

Response: Thank you very much for these observations. We made the suggested changes on page 17 as follows:

“It is worth noting that health system factors ranging from policy through primary health care delivery may influence access to effective contraceptive choices. Availability factors such as range of options, stock-outs, etc., were beyond the scope of this analysis. Contraception methods reported by WLHIV in this study were likely influenced by available options which may not be a true reflection of their preferred choice.”

Thanks once again for your review and feedback and look forward to the opportunity to publish this work in the ‘PLOS ONE’ journal. We look forward to your favourable decision. 

Sincerely,

Jim Aizire, MBChB, MHS, PhD 

Johns Hopkins Bloomberg School of Public Health 

Tel: 410-502-8988; E-mail: jaizire1@jhu.edu

---

## [Decision Letter · Decision Letter 1]

4 Aug 2023

Unintended Pregnancy and Contraception Use among African women living with HIV: baseline analysis of the multi-country US PEPFAR PROMOTE cohort

PONE-D-23-00005R1

Dear Dr. %Aizire%,

We’re pleased to inform you that your manuscript has been judged scientifically suitable for publication and will be formally accepted for publication once it meets all outstanding technical requirements.

Kind regards,

Ouma Simple

Academic Editor

PLOS ONE

Additional Editor Comments (optional):

Reviewers' comments:

Reviewer's Responses to Questions

**Comments to the Author**

1. If the authors have adequately addressed your comments raised in a previous round of review and you feel that this manuscript is now acceptable for publication, you may indicate that here to bypass the “Comments to the Author” section, enter your conflict of interest statement in the “Confidential to Editor” section, and submit your "Accept" recommendation.

Reviewer #1: All comments have been addressed

Reviewer #2: All comments have been addressed

2. Is the manuscript technically sound, and do the data support the conclusions?

Reviewer #1: Yes

Reviewer #2: Yes

3. Has the statistical analysis been performed appropriately and rigorously? 

Reviewer #1: Yes

Reviewer #2: Yes

4. Have the authors made all data underlying the findings in their manuscript fully available?

Reviewer #1: Yes

Reviewer #2: Yes

5. Is the manuscript presented in an intelligible fashion and written in standard English?

Reviewer #1: Yes

Reviewer #2: Yes

6. Review Comments to the Author

Reviewer #1: Thank you for your careful revisions and responsiveness to the reviewers. We appreciate the improvements to the paper.

Reviewer #2: the revised version resolves all of the issues from the first review. well put together study and will be helpful in future reproductive health strategies in countries involved

7. PLOS authors have the option to publish the peer review history of their article (what does this mean?). If published, this will include your full peer review and any attached files.

Reviewer #1: **Yes: **Beth A Tippett Barr

Reviewer #2: **Yes: **no

---

## [Editor Report · Acceptance letter]

28 Feb 2024

PONE-D-23-00005R1 

PLOS ONE

Dear Dr. Aizire, 

I'm pleased to inform you that your manuscript has been deemed suitable for publication in PLOS ONE. Congratulations! Your manuscript is now being handed over to our production team.

Kind regards, 

on behalf of

Dr. Ouma Simple 

Academic Editor

PLOS ONE